# Plant Genotype Shapes the Soil Nematode Community in the Rhizosphere of Tomatoes with Different Resistance to *Meloidognye incognita*

**DOI:** 10.3390/plants12071528

**Published:** 2023-04-01

**Authors:** Xiangmei Wang, Chaoyan Wang, Ru Chen, Wenxing Wang, Diandong Wang, Xueliang Tian

**Affiliations:** 1College of Biology and Food Engineering, China Three Gorges University, Yichang 443005, China; 2School of Advanced Agriculture and Bioengineering, Yangtze Normal University, Chongqing 408102, Chinaddwangwill@163.com (D.W.); 3Henan Engineering Research Center of Biological Pesticide & Fertilizer Development and Synergistic Application, Henan Institute of Science and Technology, Xinxiang 453600, China

**Keywords:** soil nematode, root exudate, tomato, cultivar, resistance

## Abstract

Soil nematodes are considered indicators of soil quality due to their immediate responses to changes in the soil environment and plants. However, little is known about the effects of plant genotypes on the soil nematode community. To elucidate this, high-throughput sequencing and gas chromatography/mass spectrometry analysis was conducted to analyze the soil nematode community and the structure of root exudates in the rhizosphere of tomatoes with different resistance to *Meloidognye incognita*. The dominant soil nematode group in the soil of resistant tomatoes was *Acrobeloides*, while the soil nematode group in the rhizosphere of the susceptible and tolerant tomatoes was *Meloidognye*. Hierarchical clustering analysis and non-metric multidimensional scaling showed that the three soil nematode communities were clustered into three groups according to the resistance level of the tomato cultivars. The soil nematode community of the resistant tomatoes had a higher maturity index and a low plant-parasite index, Wasilewska index and disease index compared to the values of the susceptible and tolerant tomatoes. Redundancy analysis revealed that the disease index and root exudates were strongly related to the soil nematode community of three tomato cultivars. Taken together, the resistance of the tomato cultivars and root exudates jointly shapes the soil nematode community. This study provided a valuable contribution to understanding the mechanism of plant genotypes shaping the soil nematode community.

## 1. Introduction

Tomato (*Lycopersicon esculentum* Mill.) is a widely cultivated vegetable throughout the world. Now, it is the most important vegetable crop in terms of production and acreage in Northern China. Along with continuous cropping for a long time, a successive cropping obstacle occurs in tomato greenhouses and results in serious soil-borne diseases, such as root rot and root-knot nematodes [1]. Specifically, root-knot nematode disease caused by *M. incognita* is the major problem of tomatoes in greenhouses [2] and needs to be controlled. Planting resistant cultivars is the most effective and economical way to control *M. incognita* [3].

In 1941, the root-knot nematode-resistant gene *Mi* was identified from *Lycopersicon peruvianum* and has shown great potential for preventing *M. incognita* [4]. Now, the *Mi* gene is a unique source of resistance in all tomato cultivars and can block nematode development at an early stage and effectively control root-knot nematode disease [3,5]. Hence, planting resistant tomatoes can suppress the population of *M. incognita* and probably changes the composition of the soil nematode community.

Soil nematodes play an important role in the decay of organic materials and nutrient transformation in the soil ecosystem. They are sensitive to agricultural management activities [6,7], nutrient states and the soil environment [8], so their populations and community structure are most vulnerable to soil disturbances. Moreover, the soil nematode community structure can measure the abundance and diversity of soil nematode assemblages and can be used to assess ecosystem stability and its capacity to sustain soil productivity and health [9]. Therefore, soil nematodes are usually regarded as indicators of soil quality. Now, high-throughput sequencing has been used as a powerful tool for revealing soil nematode communities in various ecosystems. Recently, some studies have analyzed soil nematode communities using high-throughput sequencing to reveal the structure and composition of the soil nematode communities in soil from diverse ecological zones [10,11]. 

Since soil nematodes live in close association with plant roots in the rhizosphere, the root exudates, directly and indirectly, influence soil nematodes [12]. They support the growth of microbial populations in the rhizosphere, providing a food source for microbial-feeding nematodes [13]. Furthermore, some chemical constituents of root exudates can be used by plant-parasitic nematodes to recognize and infect their plant hosts, while others repel, inhibit or even kill plant-parasitic nematodes [14,15,16]. Prior studies focused primarily on the impact of root exudates on plant-parasitic nematodes and the identification of nematocidal compounds [16,17]. In contrast, fewer studies have examined the influence of root exudates on the overall nematode community. 

Recently, resistant tomatoes have been widely planted in greenhouses in Northern China and are bound to impact the soil nematode communities. However, the mechanism of the tomato genotype shaping of the soil nematode community in the rhizosphere is unclear. Specifically, it is unclear what roles the root exudates of tomato cultivars with different resistance to *M. incognita* play in the soil nematode community. The objective was to demonstrate the effect of the plant genotype on the soil nematode communities of three tomato cultivars via the root exudates. This information will improve our understanding of the relationship between soil nematode ecology, plant genotype and plant root exudates. 

## 2. Results

### 2.1. Alpha Diversity Analysis

The numbers of soil nematode species were similar among the three soil nematode communities. Specifically, 19 species were shared by the communities of the three tomato cultivars (Figure 1A). The Simpson index of the soil nematode community of R was markedly higher than that of T and S (Figure 1B). At the genus level, the dominant nematode group in the soil nematode community of R was *Acrobeloides*, with a relative abundance of 56.6%, followed by *Meloidogyne* (30.7%). However, the main soil nematode groups in the soil nematode communities of T and S were *Meloidogyne*, with a relative abundance of 73.0% and 68.1%, respectively, and *Acrobeloides* (17.9% and 14.4%) (Figure 1C).

### 2.2. Beta Diversity Analysis

Hierarchical clustering analysis showed that the soil nematode communities of S, T and R were located in their respective branches (Figure 2A). NMDS analysis demonstrated that the soil nematode communities clustered into three groups according to the resistance level of the tomato cultivars (Figure 2B). The soil nematode community of R is located at the right part of axis 1 of the figure, and the community of S is located at the left part of axis 1 (Figure 2B), while the community of T is distributed in the lower half of the figure (Figure 2B). ANOSIM (R^2^ = 0.763, *p* < 0.001) demonstrated that significant differences existed among the three soil nematode communities (Figure 2C).

To reveal the biomarkers in the three soil nematode communities, LEfSe analysis was conducted (LDA > 3) (Figure 3A,B). The biomarkers in the community of T were OTU212 and OTU99, which belonged to *M. incognita*. In the community of S, the biomarkers were OTU 87, OTU114 and OTU 141, which were affiliated with *M. incognita*, *M. spartelensis* and a Nematoda environmental sample, respectively. The community of R had more biomarkers, such as OTU 263 and OTU270, which were assigned to *Acrobeloides nanus*, OTU251, which belonged to *Mesorhabditis* sp. and OTU252, which was identified as *Rotylenchulus reniformis*.

### 2.3. Comparison of Ecological Indices of Three Soil Nematode Communities and Disease Index

The MI of the soil nematode community of R was obviously higher than that of T and S (Figure 4A), while the PPI of the communities of T and S were markedly higher than that of R (Figure 4B). The WI of the communities of T and S were markedly lower than that of R (Figure 4C), but the NCR of the three soil nematode communities showed no significant differences (Figure 4D). The disease index of the resistant tomatoes was markedly lower than that of the susceptible and tolerant tomatoes and was consistent with the tomato resistance level (Figure 4E).

### 2.4. Effect of Root Exudates on Soil Nematode Community

The content of some of the root exudates of the three tomato cultivars was different. For example, the content of 1,3-ditert-butylbenzene, Heptadecane, *N*-eicosane, Tetradecane and 1-naphthyl aminobenzene in the root exudates of the resistant tomato cultivars were higher than those of S and T (Table 1). However, the tomato cultivars of S and T had a higher content of 2,6-ditert-butylp-cresol, Palmitamide and Oleamide (Table 1).

The RDA results (RDA1 75.95%) showed that Tetradecane (Te), 1,3-ditert-butylbenzene (Dbb), Heptadecane (He), *N*-eicosane and 1-naphthyl aminobenzene (Ne) were strongly related to the soil nematode community of R, whereas the disease index and some exudates, such as *N*-dodecane (Nd), Stearamide (St), Undecane (Un), 2,6-ditert-butylp-cresol (Dbc), Palmitamide (Pa) and Oleamide (Ol), correlated with the soil nematode communities of S and T (Figure 5A). Particularly, the disease index had the highest R^2^ value (R^2^ = 0.8674, *p* = 0.001) among the factors in the RDA results, followed by Dbc, He, Te, etc. The Mantel test also verified that the root exudates and disease index had significantly close relationships with the soil nematode communities (R = 0.569, *p* = 0.001). The two-factor correlation network demonstrated that the components of the root exudates had positive or negative influences on the soil nematode OTUs. For example, Tetradecane, 1,3-ditert-butylbenzene and Heptadecane were positively related to OTU263 (*Acrobeloides*), while 2,6-ditert-butylp-cresol, Palmitamide and Oleamide were negatively correlated with OTU263 (Figure 5B). *Meloidogyne* (OTU6, OTU99 and OTU212) was also affected by some components of the root exudates.

### 2.5. Effects of Root Exudates on M. incognita

The root exudates from the resistant tomatoes induced a higher corrected mortality rate of the J2s (19.69%) than that of the susceptible and tolerant tomatoes (7.17% and 8.66%) (Figure 6A). The suppression rate of egg-hatching by the resistant tomatoes (48.6%) was greater than that of the susceptible and tolerant tomatoes (29.8% and 28.6%) (Figure 6B). The root exudates from the resistant tomatoes exhibited repellent activity to the J2s with a marked low Cf value, while the root exudates from the susceptible tomatoes, tolerant tomatoes and sterilized water (control) attracted the *M. incognita* J2s with a high Cf value (Figure 6C).

## 3. Discussion

Our results indicated that the tomato cultivars had an impact on the alpha diversity of the soil nematode communities, resulting in differences in the Simpson index and the dominant nematode groups. The soil nematode community in the rhizosphere of the resistant tomatoes exhibited a higher Simpson index than that of the susceptible and tolerant tomato plants, indicating that the resistant tomato rhizosphere harbored more diverse soil nematodes. We speculated that the resistant tomatoes suppressed *M. incognita* and provided redundant ecological niches for other soil nematodes, thus the relative abundance of other soil nematodes increased.

The dominant soil nematode genera were different among the three communities. The main nematode group in the rhizosphere of the resistant tomatoes was *Acrobeloides*, belonging to the Ba. Our results are in line with the point that Ba is the dominant nematode in many types of soils [18,19,20]. The higher relative abundance of Ba in the soil was related to a higher population of bacteria [21]. The less Ba in the rhizosphere of the susceptible and tolerant tomatoes is probably derived from the fact that *Meloidogyne* infection reduces the bacterial population and diversity [22], thus the bacterial foods for Ba are decreased.

The dominant nematode group in the rhizosphere of the susceptible and tolerant tomatoes was *Meloidogyne* due to tomato plantation in the greenhouse for many years that promoted *Meloidogyne* accumulation in the soil. PP are the minor group in natural soil, while they can become the dominant nematode group in greenhouse soil with continuous cropping over a long time [1]. Moreover, PP directly depend on plant resources [23,24], thus the susceptible and tolerant tomato root biomass increased and provided more food for the PP, allowing them to rapidly reproduce. The smaller population of *Meloidogyne* around the resistant tomato root derived from the *Mi* gene with a strong resistance to *Meloidogyne*.

The diversity of plant genotypes can impact soil nematode communities, and lower levels of plant genotypic diversity result in a decrease in trophic-level nematodes [25], which demonstrates that plants can directly shape the soil nematode community. Moreover, plant cultivars with different resistance can influence plant-parasitic nematode abundance and diversity indices to some extent [26]. In our study, we observed distinct classifications of the three soil nematode communities, which were categorized based on the resistance level of the tomato cultivars. This reflects that the tomato genotype can significantly alter the soil nematode community structure. The observed classification was mainly attributed to the resistance of the tomato cultivars, which caused a decrease in the proportion of *Meloidogyne* and an increase in the proportion of *Acrobeloides*. Nevertheless, the susceptible and tolerant tomatoes probably played an inverse role in shaping the communities compared to the resistant tomatoes.

The biomarker taxa were the main factor contributing to the discrimination in the communities and reflected the taxonomic specificity of the community [27]. The biomarkers in the community of the resistant tomatoes were *Acrobeloides* and *Mesorhabditis*, so the soil nematode community of the resistant tomatoes was seen as a Ba community. *Acrobeloides* have short life cycles, reproduce rapidly and drive mineralization within their trophic relationship with bacteria [28], which is beneficial to the soil quality. Surprisingly, *R. reniformis*, a plant-parasitic nematode, was the biomarker in the community of the resistant tomatoes. Generally, *R. reniformis*, which has a wide range of host plants [29], is not the dominant nematode group in greenhouses. However, the increase in the *R. reniformis* population may result from the fact that resistant tomato plants are not able to inhibit the nematode. The biomarkers in the communities of S and T were *Meloidogyne* and a Nematoda environmental sample. *Meloidogyne* as a biomarker demonstrated that the communities of the susceptible and tolerant tomatoes were PP-type communities, which resulted from susceptible and tolerant host planting.

MI and PPI can be used to distinguish differences in soil nematode community structure [30] and reveal the composition of the nematode community and the degree of disturbance of the soil ecosystem [31]. Given that there were three cultivars of tomatoes planted in one greenhouse, the disturbance was not analyzed and the emphasis was on the differences in the composition of the soil nematode communities. In this study, the community of resistant tomatoes had higher MI and lower PPI than that of the susceptible and tolerant tomatoes, suggesting that more Ba and FF and less PP existed in the rhizosphere of the resistant tomatoes. Our study also showed an inverse relationship between the MI and PPI, which is consistent with previous reports that a negative correlation exists [31]. The WI may be used to assess the risk to the health of soil plants that have been infected by parasitic plants [32]. In our study, the soil nematode community in the rhizosphere of the resistant tomatoes had a higher WI, showing that the resistant tomatoes possessed a lower risk of infection by PP, which is in line with the small population of PP in the rhizosphere of the resistant tomatoes.

The content of the components of the root exudates was different in the three tomato cultivars, which is consistent with the fact that the quantity and quality of the root exudates depend on the plant species and cultivar [33]. Our RDA demonstrated that the disease index and components of the root exudates significantly shape the whole soil nematode community. Specifically, the disease index is the first contributor to the structuring of the soil nematode community, suggesting that plant resistance (consistent with the disease index) plays the main role in regulating the soil nematode community, due to the resistant tomatoes suppressing *Meloidogyne* and the susceptible and tolerant tomatoes promoting *Meloidogyne*, thus sharply changing the composition of the soil nematode communities.

It is well known that plant root exudates regulate plant–nematode interactions [34], selectively inhibit or promote some soil nematodes [17] and adjust the soil nematode community in the rhizosphere [35,36]. In our study, we found that the soil nematode communities were significantly impacted by the different components present in the root exudates. Specifically, we observed that various components of the exudates simultaneously suppressed or promoted each soil nematode species. This suggests that the effect of the root exudates on soil nematodes is a comprehensive outcome that relies on various components working together. To verify the effect of the root exudates on the soil nematodes, we used the crude root exudates as a replacement for the chemical component to assess the influence on *Meloidognye*. The crude root exudates of the resistant tomatoes suppressed *Meloidognye*, while the exudates of the susceptible and tolerant tomatoes improved *Meloidognye*, which is supported by previous reports that the root exudates of resistant and susceptible plants can directly repel or promote plant-parasitic nematodes [12,34,37,38]. In terms of Ba, plant exudates may supply a source of organic carbon to soil microbes in the rhizosphere, thus providing many food sources for Ba and stimulating a Ba increase, finally promoting the Ba population. It is assumed that the tomato resistance (genotype) combined with the root exudates jointly constructed the soil nematode communities.

This study illustrated that the soil nematode communities are shaped by tomato resistance to *M. incognita* and root exudates using a high-throughput sequencing method. Nevertheless, due to incomplete public sequence databases for nematodes, many amplicon sequences were only categorized as ‘Nematoda environmental sample’ rather than a specific species. Consequently, several rare species may have been missed. As more soil nematode sequences become available, the precision of the molecular taxonomy is predicted to be enhanced, resulting in the discovery of more elusive species.

## 4. Materials and Methods

### 4.1. Tomato and M. incognita Materials

The three tomato cultivars included maker, 888 and Xianke for this experiment. The money maker cultivar is a susceptible cultivar with susceptibility to *M. incognita*. The 888 cultivar is a tolerant cultivar with tolerance to *M. incognita* and is widely planted in local greenhouses. The Xianke cultivar, carrying the *Mi* gene, is a resistant cultivar with resistance to *M. incognita*. 

The *M. incognita* used in this study was obtained from the Henan Institute of Science and Technology. The second stage *M. incognita* juveniles (J2s) were inoculated on pepper (*Capsicum annuum* cv. Qiemen) plants and maintained in the greenhouses for 40 days until egg masses formed. The egg masses were treated with 2% NaOCl and incubated at 28 °C for 24 h in sterile water to hatch the J2s.

The tomato seeds were surface-sterilized in 0.5% NaOCl, washed thoroughly with sterile water and germinated on sterile, moist filter paper in a Petri dish for 3 days at 28 °C under darkness. Then, the germinated tomato seeds were maintained in a sterile mixture of peat:vermiculite (2:1, *vol/vol*) in a growth chamber with a 15 h light and 9 h dark cycle at 28 °C for 2 weeks to obtain tomato seedlings. For the field experiment, 50 seedlings of each tomato cultivar were planted in a greenhouse. For the exudate collection, 30 seedlings of each tomato cultivar were planted in 10 pots (3 plants per pot) and inoculated with about 500 J2s in each pot. At 40 days post-inoculation, root exudates obtained from the roots of three tomato cultivars were used for GC-MS analysis. 

### 4.2. Collection of Root Exudates and Gas Chromatography/Mass Spectrometry Analysis

The root exudates were collected according to a previous method [39] with slight modifications. First, the tomato seedlings of each cultivar were carefully removed from the chamber and washed gently with sterilized distilled water to remove the nutrition medium. They were classified into three groups (10 plants per group) and seen as three replicates. The three groups of tomato seedlings were dipped into 500 mL of sterilized distilled water in a 1000 mL beaker, respectively. The beaker was shaded with silver paper to maintain the roots in the dark. Moreover, the sterilized distilled water was aerated for root respiration. The tomato seedlings in the beaker were maintained under a temperature range of 23 to 25 °C. After 24 h, the distilled water was collected and filtered with a bacteria filter to remove the root residue. A total of 50 mL of the filtrate was used to perform the bioassay, and the residues were concentrated to 20 mL with a vacuum rotary evaporator. The concentration procedure was conducted according to Yang’s method [37]. The chemical constituents of the root exudates were detected by gas chromatography/mass spectrometry (GC-MS) (Agilent, Santa Clara, CA, USA, 6890-5973N). The procedure and conditions were conducted according to the method of Yang et al. (2016).

### 4.3. Tomato Planting in the Greenhouse

The greenhouse was located in Xinxiang (latitude 35°18′ N, longitude 113°52′ E), Henan Province, China. The greenhouse had already been used for the planting of cucumbers and tomatoes for 20 years. The disease caused by *M. incognita* was heavy in this greenhouse. The tomato seedlings at the 4-leaf stage were transplanted in the greenhouse on 3 April. Fifty tomato seedlings for each cultivar were arranged at intervals on 5 ridges in the greenhouse. Each ridge served as one repeat, thus each cultivar had five repeats. The common methods of cultivation and management were conducted. When the tomatoes had grown for 2 months, we collected the rhizosphere soil, as follows: the plants were removed from the soil, and the soil was shaken off and regarded as the rhizosphere soil. The soil was put into plastic bags for the isolation of soil nematodes. The soil samples of money maker, 888 and Xianke were named S, T and R, respectively. The disease index of *M. incognita* was investigated as follows [40]. A root gall index (1–5) was calculated as follows: 1, no galls; 2, 1–25% of roots with galls; 3, 26–50% with galls; 4, 51–75% with galls; and 5, >75% with galls. 

### 4.4. Soil Nematode Isolation and DNA Extraction

Soil nematodes were isolated from 100 g of fresh soil per sample using the modified Baermann funnel method [41]. The extraction of the soil nematodes was conducted three times from each soil sample and mixed together. The soil nematodes were finally collected into 1.5 mL centrifuge tubes for DNA extraction. For each tomato cultivar, five tubes of soil nematodes were collected as five repeats. To avoid the negative effect of humic acid on the soil nematode DNA, the DNA was extracted using the MOBIO UltraClean™ Soil DNA Isolation Kit (Mo Bio Laboratories, Carlsbad, CA, USA) according to the manufacturer’s instructions. The DNA concentration and purity were assessed on 1% *w/v* agarose gel. 

### 4.5. PCR Amplification and Pyrosequencing

The primers NF1(5′-GGT GGT GCA TGG CCG TTC TTA GTT-3′) [42] and 18Sr2b (5′-AGC GAC GGG CGG TGT GTA CAA A-3′) [43] were used in this study to obtain amplicons for MiSeq pyrosequencing. The PCR procedure was as follows: predenaturation at 95 °C for 3 min, 30 cycles including denaturation at 95 °C for 30 s, annealing at 55 °C for 30 s and extension at 72 °C for 45 s, 72 °C for 5 min (one cycle). The PCR reactions were performed in triplicate of a 20 μL mixture containing 4 μL 5 × FastPfu buffer, 2 μL 2.5 mM dNTPs, 0.8 μL forward primer (5 μM), 0.8 μL reverse primer (5 μM), 0.4 μL FastPfu polymerase and 10 ng of template DNA. The purified amplicons were pooled and paired-end sequenced (2 × 300) on an Illumina MiSeq platform (Illumina, San Diego, CA, USA) according to the standard protocols by Majorbio Bio-Pharm Technology Co., Ltd. (Shanghai, China).

### 4.6. Bioinformation Analysis

Bioinformatics analysis was conducted on the free online Majorbio I-Sanger Cloud Platform (https://www.i-sanger.com/ (accessed on 4 September 2021)). The raw sequences were processed using Quantitative Insights Into Microbial Ecology software (1.7.0) [44]. The low-quality sequences, such as the read mismatching sequences, sequences shorter than 50 bp, PCR-based or sequencing errors and chimeras, were removed. The high-quality sequences were used to identify the operational taxonomic units (OTUs) at 97% similarity by Uclust [45]. The represented sequence of each OTU was aligned to the reference set of nematode sequences [46], and non-nematode sequences, including fungi, algae and plants, were removed. 

For Alpha diversity, the number of OTUs and the Simpson index were used to evaluate the diversity of the soil nematode community. The composition of the soil nematode community was assessed at a genus level. For Beta diversity, a hierarchical cluster dendrogram, non-metric multidimensional scaling (NMDS) analysis and analysis of similarities (ANOSIM) were performed to determine the differences in the soil nematode communities [47,48,49]. The biomarker genera among the three soil nematode communities were identified by Linear Discriminant Analysis Effect Size (LEfSe) [50]. The redundancy analysis (RDA) in the R package vegan was used to detect the effect of the root exudates and disease index on the soil nematode community across three cultivars [51]. Mantel test analysis was run to reveal the relationship between the root exudates, disease index and soil nematode communities [52]. A two-factor correlation network was conducted to investigate the relationships between the soil nematodes and the components of the root exudates based on Spearman correlations and to build the correlation network on the Majorbio platform with default parameter settings (https://cloud.majorbio.com/ (accessed on 4 September 2021)). 

### 4.7. Ecological Index Analysis

Based on OTU taxonomy, the soil nematodes were classified into bacterivores (Ba), fungivores (Fu), omnivores/predators (OP) and plant parasites (PP). To analyze the ecological index of the soil nematode communities, some community indices were calculated: the maturity index of free-living nematodes (MI). MI = ∑v(i)f(i), where v(i) is the c-*p* value of non-plant-parasitic nematodes i and f(i) is the frequency of that taxon in a sample; the maturity index of plant-parasitic nematodes (PPI). PPI = ∑v(i)f’(i), where v(i) is the c-p value of the plant-parasitic nematodes i and f’(i) is the frequency of the nematodes in a sample [53,54]; the Wasilewska index (WI). WI = (B + F)/P, where B, F and P were the proportions of Ba, Fu and PP in the soil nematode communities [32]; and the nematode channel ratio (NCR). NCR = B/(B+F), where B and F were the proportions of Ba and Fu in the soil nematode communities [55].

### 4.8. Effect of Root Exudates on Mortality Rate of M. incognita J2

About 100 J2s in 50 μL of nematode suspension were placed in a well with 950 μL of root exudate and incubated in an incubator at 25 °C. The dead J2 were determined by the NaOH method [37,56] and counted under a microscope at 24 h. The sterilized water served as a negative control. The experiment was repeated three times. The corrected mortality rate was calculated according to the following formula: corrected mortality rate (%) = (mortality rate of J2 in root exudate treatment − mortality rate of J2 in control)/(1 − mortality rate of J2 in control) × 100%.

### 4.9. Effect of Root Exudates on M. incognita Egg-Hatching

A total of 150 *M. incognita* eggs were mixed with 500 μL root exudate in separate wells. The 24-well plates were incubated at 25 °C for 72 h in a growth chamber. The sterilized water served as a negative control. The experiment was repeated three times. The numbers of hatched J2s were counted at 72 h after incubation. The suppression of the egg-hatching rate was calculated according to the following formula [37]: suppression rate (%) = (number of J2 in control − number of J2 in root exudate)/number of J2 in control × 100.

### 4.10. Effect of Root Exudates on the Chemotaxis of M. incognita J2

The chemotactic activity of tomato exudates towards J2, as well as their capacity to either attract or repel nematodes, was tested according to the previously described method [16]. First, a 5 cm petri dish was divided into 16 segments and filled with 5 mL water agar at a concentration of 0.4%. After the culture medium was cooled, two filter paper pieces (1 cm diameter) were dipped in tomato root exudates and placed on the two sides of the dish. After 1 h, 50 μL suspensions containing about 100 J2s were dropped onto the medium in the middle of the plate. The dish was incubated at 25 °C for 3 h in the dark. The J2s can move on the medium due to being attracted or repelled by the root exudates. Afterwards, 70% ethanol was sprayed on the medium to stop the movement of the J2s, and the number of J2s on each segment was recorded under a stereomicroscope. The sterilized water served as a negative control. Each treatment contained 10 replicates. The grid of 16 segments on the medium was designated by the labels (1), (2)…(8) and (a), (b)…(h). Each segment was then assessed for the presence (1) or absence (0) of nematode tracks. The chemotaxis was evaluated according to the following formula: the chemotaxis factor (Cf) = affinity/repellence = [scores(1) + (2) + … + (8)]/[scores(a) + (b) + … + (h)].

A Cf greater than 2 meant the attraction of nematodes, while a value lower than 0.5 meant repellence.

### 4.11. Data Analysis

The differences in the content of the chemical components of the root exudates, the effect of the root exudates on *Meloidogyne* (corrected mortality rate, suppression rate of egg-hatching rate, chemotaxis), the ecological indices and disease index of the soil nematode communities among the three tomato cultivars were tested by ANOVA.

## Figures and Tables

**Figure 1 plants-12-01528-f001:**
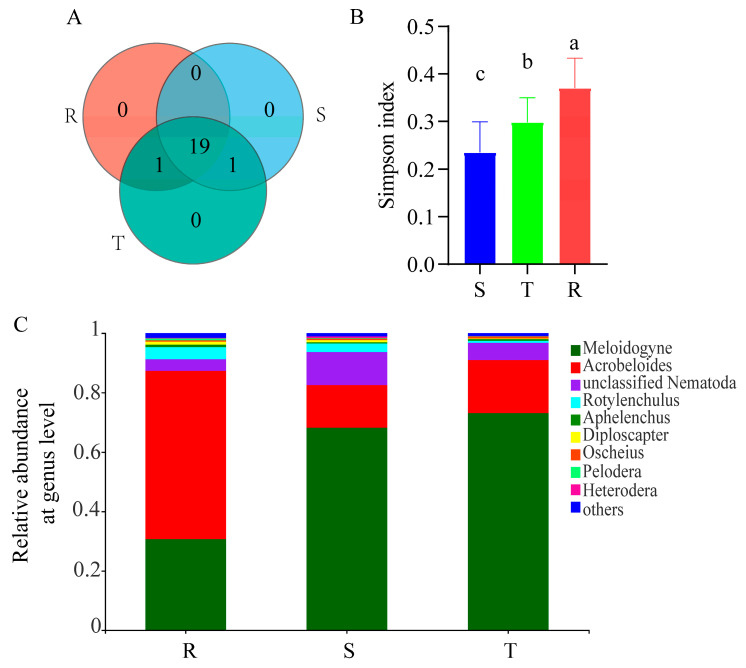
Alpha diversity of soil nematode community. (**A**), Number of species. (**B**), Simpson index. (**C**), Composition of soil nematode community at genus level. Different letters between the columns show the significant difference between the three tomato cultivars. S: susceptible cultivar. T: tolerant cultivar. R: resistant cultivar.

**Figure 2 plants-12-01528-f002:**
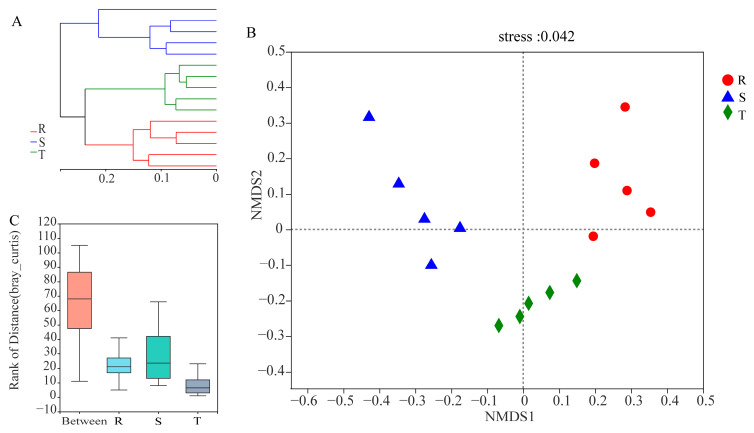
Beta diversity analysis of three soil nematode communities. (**A**), Hierarchical clustering analysis. (**B**), NMDS. (**C**), ANOSIM. S: susceptible cultivar. T: tolerant cultivar. R: resistant cultivar.

**Figure 3 plants-12-01528-f003:**
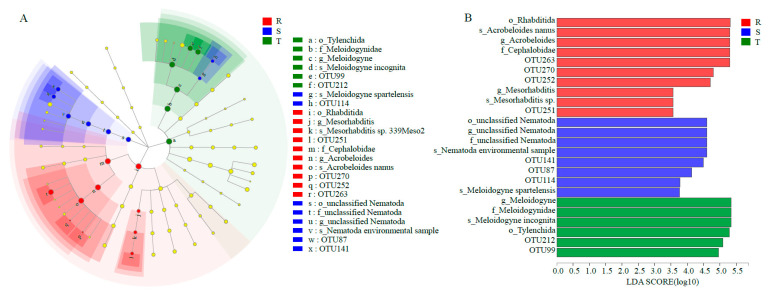
LEfSe analysis of three soil nematode communities. (**A**), LefSe analysis. The cladogram shows the taxa with marked differences in the three soil nematode communities. Red, green and blue indicate different groups, with the classification of taxa at the level of class, order, family and genus shown from the inside to the outside. (**B**), Species with significant differences that had an LDA score higher than the estimated value; the default score is 3.0. The length of the histogram represents the LDA score. S: susceptible cultivar. T: tolerant cultivar. R: resistant cultivar.

**Figure 4 plants-12-01528-f004:**
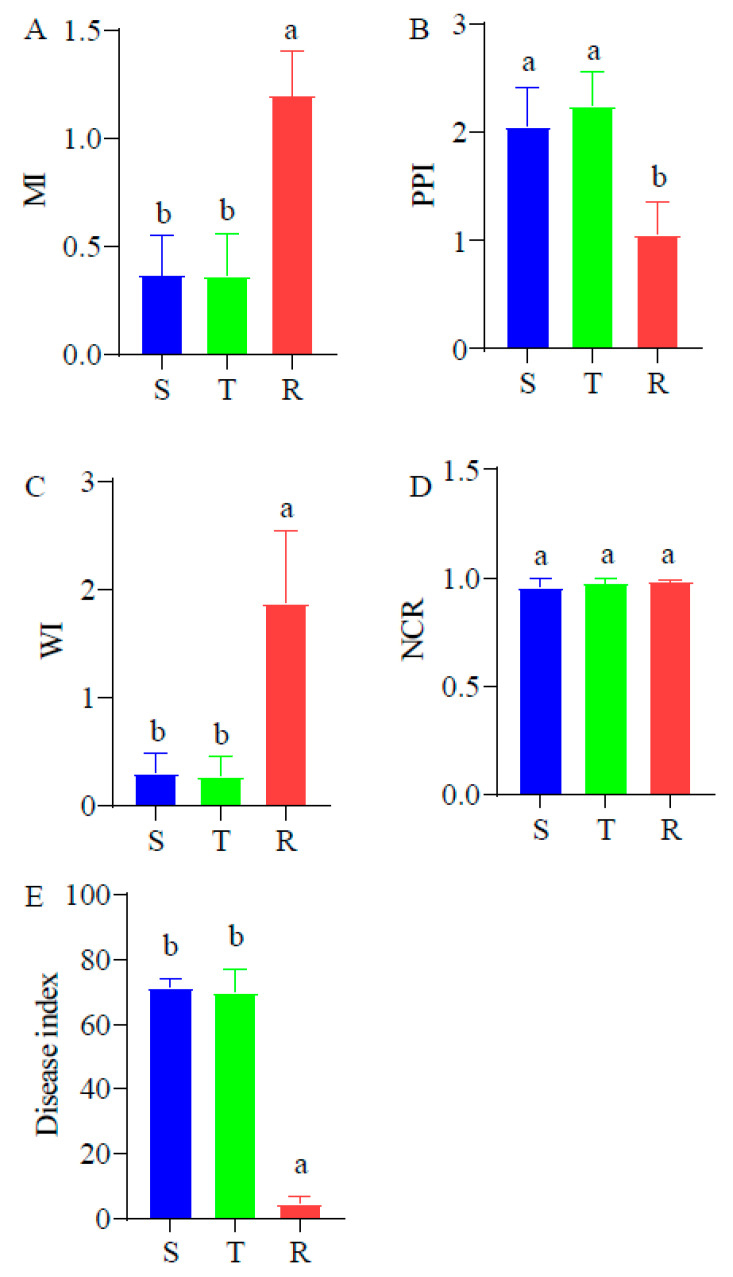
Difference in ecological indices of three soil nematode communities. (**A**), MI. (**B**), PPI. (**C**), WI. (**D**), NCR. (**E**), Disease index. Different letters between the columns show the significant difference among the three soil nematode communities. S: susceptible cultivar. T: tolerant cultivar; R: resistant cultivar.

**Figure 5 plants-12-01528-f005:**
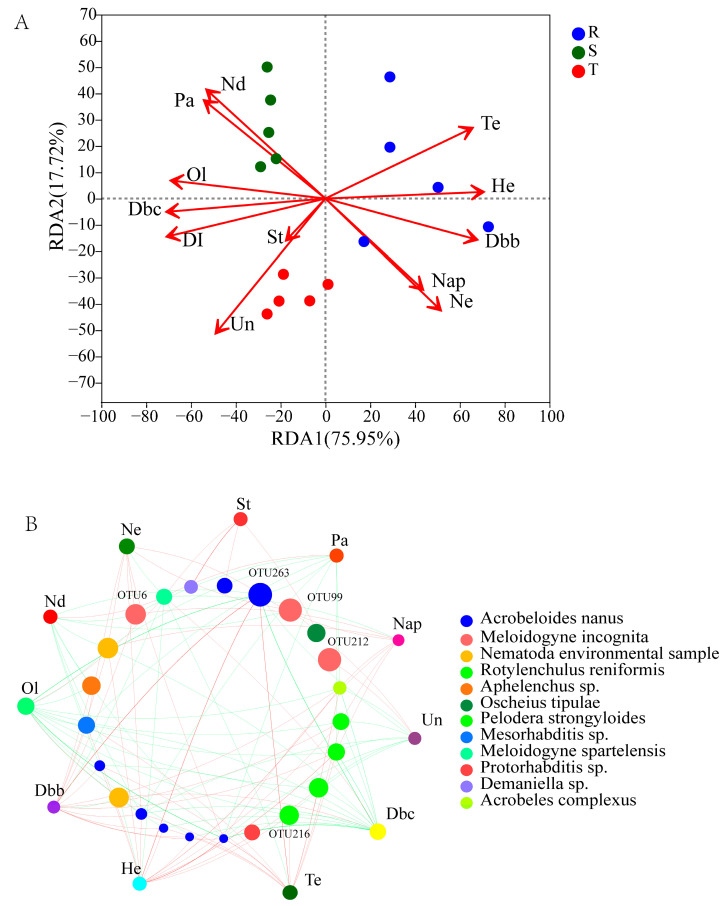
RDA and two-factor correlation network revealed the influence of root exudates on soil nematode communities. (**A**), RDA. (**B**), Two-factor correlation network. Red line represents positive correlation and green line represents negative correlation. Dbb: 1,3-ditert-butylbenzene; Dbc: 2,6-ditert-butylp-cresol; He: Heptadecane; Nap: 1-naphthyl aminobenzene; Nd: *N*-dodecane; Ne: *N*-eicosane; Ol: Oleamide; Pa: Palmitamide; St: Stearamide; Te: Tetradecane; Un: Undecane. S: susceptible cultivar. T: tolerant cultivar. R: resistant cultivar. DI: disease index.

**Figure 6 plants-12-01528-f006:**
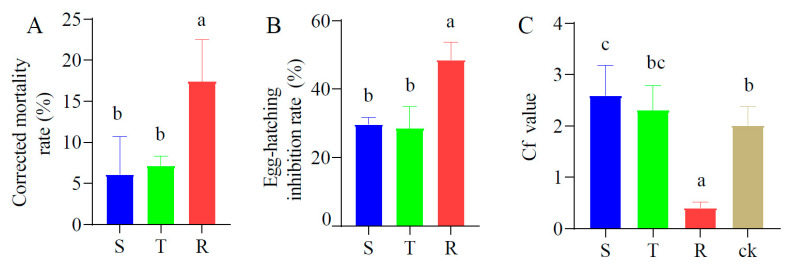
Effects of root exudates from three tomato cultivars on *M. incognita*. (**A**), Corrected mortality rate. (**B**), Suppression rate of egg-hatching. (**C**), Chemotaxis. Different letters on the column show the significant differences among three tomato cultivars. S: susceptible cultivar. T: tolerant cultivar. R: resistant cultivar.

**Table 1 plants-12-01528-t001:** The root exudates with significant differences between the three tomato cultivars.

Root Exudates	Susceptible Cultivar	Tolerant Cultivar	Resistant Cultivar
Tetradecane	7.28 ± 0.07 a	6.85 ± 0.08 b	8.86 ± 0.05 c
*N*-dodecane	4.94 ± 0.05 a	4.41 ± 0.04 b	4.31 ± 0.05 c
1,3-ditert-butylbenzene	1.61 ± 0.02 a	2.26 ± 0.02 b	3.06 ± 0.06 c
Heptadecane	3.06 ± 0.05 a	3.63 ± 0.03 b	5.85 ± 0.06 c
*N*-eicosane	9.75 ± 0.04 a	12.20 ± 0.09 b	12.60 ± 0.03 c
2,6-ditert-butylp-cresol	16.44 ± 0.04 a	16.18 ± 0.05 b	14.94 ± 0.05 c
Palmitamide	4.78 ± 0.04 a	4.14 ± 0.03 b	3.96 ± 0.02 c
Oleamide	26.62 ± 0.14 a	25.15 ± 0.05 b	22.29 ± 0.08 c
1-naphthyl aminobenzene	1.27 ± 0.03 a	1.47 ± 0.03 b	1.50 ± 0.05 b
Stearamide	4.05 ± 0.02 a	4.27 ± 0.79 a	3.73 ± 0.05 a
Undecane	2.49 ± 0.05 a	2.91 ± 0.03 b	2.20 ± 0.04 c

Data represent the mean ± standard error of the three independent biological replicates and was analyzed by one-way ANOVA with Duncan’s multiple range test. Different letters indicate significant differences (*p* < 0.05). S: susceptible cultivar. T: tolerant cultivar. R: resistant cultivar.

## Data Availability

The datasets used or analyzed during the study are available from the corresponding author upon reasonable request. The raw reads of the ITS MiSeq data were deposited in the NCBI Sequence Read Archive database (PRJNA898310).

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
