# Peer review of "Plant Genotype Shapes the Soil Nematode Community in the Rhizosphere of Tomatoes with Different Resistance to Meloidognye incognita"

_plants, 2023, doi:10.3390/plants12071528_

Round 1

Reviewer 1 Report

Please check: 

Line 52 – ecological soils – what this means?

Line 258 - 4. Discussion – should go one line bellow 

Author Response

  1. Line 52 – ecological soils – what this means?

Reply:

We changed the sentence to ‘soil samples taken from diverse ecological zones’. (line 46)

  1. Line 258 -4. Discussion – should go one line bellow 

Reply:

We revised it.

Reviewer 2 Report

I think that the article is interesting and quite novel. The experiment is well design and the results are too clear that suprise me. The three tomato cultivars are clearly distinguished in terms of root exudates and the composition of the rhizosphere nematode community. The authors attribute these differences to the resistance of plants sto M.incognita. Are there differences in other characteristics between the cultivars that could possibe induce the above mentioned differences? For instance differences in growth rates or leaf characteristics?

The parts that needs serious revisions is the introductio that is too short and discussion that needs considerable linguistic revisions. In some points I couldn't understand the meaning of the text.

I provide an annotated pdf where the authors could find my detailed remarks

Author Response

We greatly appreciate the valuable feedback and corrections provided by the reviewer, who dedicated his time and effort to improve our manuscript.

Reviewer: 2

I think that the article is interesting and quite novel. The experiment is well design and the results are too clear that surprise me. The three tomato cultivars are clearly distinguished in terms of root exudates and the composition of the rhizosphere nematode community. The authors attribute these differences to the resistance of plants to M.incognita.

  1. Are there differences in other characteristics between the cultivars that could possible induce the above mentioned differences? For instance differences in growth rates or leaf characteristics?

Reply:

The soil nematodes residing in the plant rhizosphere are highly likely to be influenced by root exudates. Thus, in this study, we compared the root exudates of three different tomato cultivars. As the growth rates and leaf characteristics are not directly linked to soil nematodes, we did not analyze these parameters.

  1. The parts that needs serious revisions is the introduction that is too short and discussion that needs considerable linguistic revisions. In some points I couldn't understand the meaning of the text.

Reply:

We add one paragraph about root exudates in the part of introduction (line 48-55).

Since soil nematode live in close association with plant roots in the rhizosphere, root exudates directly and indirectly influence soil nematodes [12]. They support the growth of microbial populations in the rhizosphere, providing a food source for microbial-feeding nematodes [13]. Furthermore, some chemical constituents of root exudates can be used by the plant-parasitic nematodes to recognize and infect their plant hosts, while others repel, inhibit, or even kill plant-parasitic nematodes [14-16]. Prior studies focused primarily on the impact of root exudates on plant-parasitic nematodes and on identifying nematocidal compounds [16,17]. In contrast, fewer studies have examined the influence of root exudates on soil nematodes themselves.

  1. Line 11 this analysis was used to identify the structure of root exudates not of nematodes. the authors must mention this in the abstract

Reply:

We added ‘and the structure of root exudates’ in line 8.

  1. Line 21 are tomato cultivars and root exudates or tomato cultivars through root exudates shape the nematode community?

Reply:

We think that both tomato resistance and root exudates play a role in shaping the nematode community. The resistant tomato can directly inhibit M.incognita infection and reduce the abundance of PP. In addition, the root exudates of resistant tomato also indirectly influence soil nematodes. On the other hand, the susceptible and tolerant tomato, along with their root exudates, can promote PP.

  1. Line 65-66 which is the difference between tolerance and resistance to M. incognita?

Reply:

The tolerant tomato cultivar is susceptible to M. incognita, which leads to the formation of root knots on its roots, but its yield does not decrease significantly due to its high compensation ability.

  1. Line 86 they grouped in relation to which criterion? they were 30 seedlings in 10 pots and then they were divided in 3 groups of 10 seedlings each one? If this was the procedure why this happened?

Reply:

The quantity of root exudates is quite minimal. To collect more root exudates, 30 seedlings were used for each cultivar. The 30 seedlings were divided into three groups for three replicates, with each replicate containing 10 seedlings. To ensure proper growth of the seedlings, three of them were grown in a single pot.

  1. Line 98 for the determination of root exudates, seedlings in pots into the greenhouse were used while for the determination of nematode community seedlings in greenhouse was used?

Reply:

Thirty seedlings of each tomato cultivar were grown in pots to collect exudates, while another fifty seedlings of each cultivar were planted in a greenhouse and analyzed for soil nematode communities after two months of growth.

  1. Line 112 5 samples per treatment?

Reply:

Fifty tomato seedlings for each cultivar were arranged at interval on 5 ridges in a greenhouse. Each ridge was served as one repeat, thus each cultivar had five repeats. Each repeat contains 3 soil samples. Therefore, each treatment (cultivar) had a total of 15 samples, and the extraction of soil nematodes was performed 15 times for each treatment.

  1. Line 113 Is this the common procedure?

Reply:

To minimize potential errors in soil sampling, we followed the procedure outlined in our previous paper (Tian et al., 2020) and combined three soil nematode samples.

Xueliang Tian, Xiaoman Zhao, Zhenchuan Mao, Bingyan Xie. Variation and Dynamics of Soil Nematode Communities in Greenhouses with Different Continuous Cropping Periods. Horticultural Plant Journal, 2020, 6 (5): 301–312.

  1. Line 115 why the authors prefer to use molecular techniques to identify nematodes and not the usual way by microscope. I think that the second way has a better discriminatory power since the molecular identification depends on the available nematode genomes that are not so many.

Reply:

Yes. We agree with reviewer’s point.

Darby et al. (2013) proposed that the completeness of public sequence databases would promote accuracy of amplicon sequencing applied to the soil nematode community. To date, the public databases contained more sequences of soil nematode and can reveal the soil nematode by molecular method to some extent. Some studies have analyzed soil nematode communities by high-throughput sequencing to reveal the structure and composition of the soil nematode communities in different soils (Kenmotsu et al, 2021a; Kenmotsu et al, 2021b; Sapkota and Nicolaisen, 2015 ).

The microscopy method has drawbacks, such as requiring inefficient morphology-based analyses, and it would have been difficult to analyze the numerous samples in our study. Consequently, we chose to adopt molecular techniques which made the study of soil nematodes easier and more efficient.

Darby, B., Todd, T.C., Herman, M.A. High-throughput amplicon sequencing of rRNA genes requires a copy number correction to accurately reflect the effects of management practices on soil nematode community structure. Mol Ecol, 2013, 22: 5456–5471 .

Sapkota R, Nicolaisen M. High-throughput sequencing of nematode communities from total soil DNA extractions. BMC Ecol. 2015,15:3.

Kenmotsu H, Takabayashi E, Takase A, Hirose Y, Eki T. Use of universal primers for the 18S ribosomal RNA gene and whole soil DNAs to reveal the taxonomic structures of soil nematodes by high-throughput amplicon sequencing. PLoS One. 2021b,16(11):e0259842.

Kenmotsu H, Ishikawa M, Nitta T, Hirose Y, Eki T. Distinct community structures of soil nematodes from three ecologically different sites revealed by high-throughput amplicon sequencing of four 18S ribosomal RNA gene regions. PLoS One. 2021a, 15;16(4):e0249571.

  1. Line 140 species or OTUs?

Reply:

OTUs. We revised it.

  1. Line 181 give some details about the method. what it measures?

Reply:

We revised the sentence as followed ‘The chemotactic activity of tomato exudates towards J2, as well as their capacity to either attract or repel the nematodes,’ (line 179-180)

  1. Line 192 what is 1 to 8 and a to h in the formula?

Reply:

We added the following sentence to the text (line 189-191)

The grid of 16 segments on the medium was designated by the labels (1), (2)…(8) and (a), (b)…(h). Each segment was then assessed for the presence (1) or absence (0) of nematode tracks.

  1. Line 221 what is this?

Reply:

The Nematoda environmental sample refers to the taxonomy of nematode Operational Taxonomic Units (OTUs) identified in the NCBI database. These OTUs were not assigned to known nematode species, and as a result, they were named ‘Nematoda environmental sample’.

  1. Line 222 A. nanus please write the full name

Reply:

We revised it. ‘A. nanus’ was changed to ‘Acrobeloides nanus’

  1. Line 237-241 the legend is missing from this figure and from other figures as well. So I cannot find the correspondence between the acronyms presented in the Figure and the names of the root exudates or disease index or whatever.

Reply:

We added the acronyms in the text (line 238-242).

  1. Line 243-244 where are the results of the Mantel test?

Reply:

Mantel test was used to calculate the correlation between soil nematode and root exudates, the value of R > 0.5 represented they had close correlation. In our study, ‘R=0.569, P=0.001’ (line 245) was the result of mantel test.

  1. Line 245-246 is this based on spearman correlations? Some more details for the analysis used be resented on data analysis section

Reply:

Two-factor network analysis can calculate the correlation between species and environmental factors based on Spearman correlations, build the correlation network, and analyze the correlation between species and environmental factors.

We added one sentence ‘based on Spearman correlations and to build the correlation network’ in text (line 150)

  1. Line 249 how could the authors see whether the relation is positive or negative?

Reply:

In Figure 5B, Two-factor network showed that red line represented positive correlation and green line represented negative correlation.

  1. Line 258 why you present the values of control only for Cf and not for mortality and % hatching eggs?

Reply:

We corrected the formula of suppression of egg-hatching rate: suppression rate (%) = (number of J2 in sterilized water-number of J2 in root exudate) / number of J2 in sterilized water × 100(line 175-176)

The corrected mortality rate and suppression of egg-hatching rate were calculated by the values of control, thus the control was not presented in the figure.

  1. Line 259 not clustered. Maybe organized or structured or distinguished

Reply:

We deleted the sentence and added one sentence ‘Our results showed that tomato cultivar affected the alpha diversity of the soil nematode communities and caused different Simpson index and dominant nematode group’(line 260-261)

  1. Line 261 what do you mean by these?

Reply: we also thought the sentence was redundant, so deleted the sentence.

  1. Line 263 please cane te syntax of this sentence and other similar sentence as well. the right expression is "the nematode community in the rhizosphere of the resistant tomato exhibited higher....

Reply:

We revised the sentence ‘The soil nematode community in the rhizosphere of the resistant tomato exhibited higher Simpson index than that of susceptible and tolerant tomato plants, indicating that the resistant tomato rhizosphere harboured more diverse soil nematodes.’ (line 265-268)

  1. Line 264 why the higher Simpson means higher stability?

Reply:

We revised the sentence ‘The soil nematode community in the rhizosphere of the resistant tomato exhibited higher Simpson index than that of susceptible and tolerant tomato plants, indicating that the resistant tomato rhizosphere harboured more diverse soil nematodes’. (line 265-268)

  1. Line 267-268 please rephrase to be more understandable

Reply:

We deleted the sentence due to it did not sustain our point.

  1. Line 270 nematode groups?genera?

Reply:

It should be genera. We revised it.

  1. Line 272 in all type of soils? even in forest?

Reply:

We revised the sentence and added one reference. (line 277)

Liu, X., Zhang, D., Li, H., Qi, X., Gao, Y., Zhang, Y., Han, Y., Jiang, Y., Li, H. Soil nematode community and crop productivity in response to 5-year biochar and manure addition to yellow cinnamon soil. 2020. BMC Ecol,20, 39.

  1. Line 279 I think that PP is a very common group in agricultural soils and not a minor group as the authors said

Reply:

We deleted ‘ or agricultural’.

  1. Line 281 the same holds in agricultural areas

Reply:

Yes. We agree with you.

  1. Line 282 do you mean that the root biomass in S and T is higher than in R? I think that the major driver of changes is the quality of exudates than their amount

Reply:

We did not measure the root biomass of the three tomato cultivars. PP can feed on susceptible and tolerant tomato cultivars and reproduce rapidly. However, PP is unable to infect the roots of resistant tomato plants and is also inhibited by root exudates of resistant tomato plants. As a result, their population decreases.

  1. Line 284 what exactly this gene does? How it affect M.incognita?

Reply:

The Mi gene was a root-knot nematode resistance gene and a unique source of resistance in all tomato cultivars, able to block nematode development at an early stage.

  1. Line 285-286 what do you mean by plant genotype diversity? please rephrase. I can not get it.

Reply:

We changed the sentence to ‘The diversity of plant genotypes can impact soil nematode communities, and lower levels of plant genotypic diversity result in a decrease in trophic-level nematodes’. (line 290-291)

The sentence ‘due to complementarity effects of multiple plant species [41],’ was deleted.

  1. Line 290 ou have already mentioned this at the first sentence of the discussion.

Reply:

We changed the first sentence of the discussion to ‘Our results indicated that the tomato cultivar had an impact on the alpha diversity of soil nematode communities, resulting in varying Simpson indices and dominant nematode groups ’. (Line 260-261) .

The sentence (Line 290 in review manuscript) was also revised as followed.

In our study, we observed a distinct classification of three soil nematode communities, which were categorized based on the resistance level of the tomato cultivar. This reflects that the tomato genotype can significantly alter the soil nematode community structure.

  1. Line 292 see my previous comment

Reply:

The sentence was revised as followed.

The observed classification was mainly attributed to the resistance of the tomato cultivar, which caused a decrease in the proportion of Meloidogyne and an increase in the proportion of Acrobeloides.

  1. Line 302-305 rephrase to be understandable

Reply:

We revised the sentence.

Generally, Rotylenchulus reniformis, which has a wide range of host plants [49], is not the dominant nematode group in greenhouses. However, the increase in R. reniformis population may result from the fact that resistant tomato plants are not able to inhibit the nematode. (line 313-315)

  1. Line 308-310 this could be mentioned as a limitation of this research at the end of the discussion

Reply:

We added a conclusion section as followed.

The study illustrated that the soil nematode communities are shaped by tomato resistance and root exudates using a high-throughput sequencing method. Nevertheless, due to the incomplete public sequence databases for nematodes, many amplicon sequences were only categorized as ‘Nematoda environmental sample’ rather than a specific species. Consequently, several rare species may have been missed. As more soil nematode sequences become available, the precision of molecular taxonomy is predicted to enhance, resulting in the discovery of more elusive species. (line 362-367)

  1. Line 317 I think that higher MI is related to bacteria and fungi with higher cp values

Reply:

The higher MI was derived from high abundance of Ba and FF and low abundance of PP.

  1. Line 318 why the authors refer only to Ba and not to Fu nematodes that participate in MI?

Reply:

We change the sentence to ‘more Ba and FF, less PP’.

  1. Line 336-339 these are results

Reply:

The sentence was changed to ‘ In our study, we found that the soil nematode communities were significantly impacted by the different components present in root exudates. Specifically, we observed that various components of the exudate simultaneously suppressed or promoted each soil nematode species. This suggests that the effect of root exudates on soil nematodes is a comprehensive outcome that relies on various components working together’. (line 344-348)

  1. Line 340 I cannot get it

Reply:

The two-factor correlation network demonstrated that different components of root exudate could simultaneously suppress or promote each soil nematode species. Similarly, each component of root exudate could also influence numerous species of soil nematodes. In the soil environment, root exudates contain compounds with multiple components, meaning that their effects on soil nematodes are attributable to various types of components.

We added one sentence ‘To verify the effect of root exudates on soil nematode, ’ to text (line 352).

  1. Line 341 did this happen in this study?

Reply:

We did not obtain each component of root exudates, so we used the crude root exudates to test the influence on Meloidognye.

  1. Line 347 rephrase

Reply:

We changed the sentence to ‘foods for Ba living and stimulate Ba increasing’.

Furthermore, some of the grammar mistakes noted by the reviewer have been revised accordingly. The specific revisions can be found within the text with revised trace.

Round 2

Reviewer 2 Report

The authors have been replied to the comments in details and have incorporated most of the suggestions.However, I recommend  some minor linguistic suggestions that I mention in the annotated pdf.

Author Response

The authors have been replied to the comments in details and have incorporated most of the suggestions. However, I recommend some minor linguistic suggestions that I mention in the annotated pdf.

  1. in both formulas use the same term (control or sterilized water) (line 199)

Reply:

The ‘sterilized water’ were changed to ‘control’.(line 176-177)

  1. what do you mean by playing an inverse role? (line 332)

Reply:

Yes. The susceptible and tolerant varieties of tomato plant supported and encouraged PP, whereas the resistant variety suppressed it. As a result, these tomato varieties play opposite roles in influencing the community of soil nematodes.

  1. at this sentence the verb is missing(line 361)

Reply:

We revised the sentence as follow. The WI may be used to assess the risk to the health of soil plants that have been infected by parasitic plants

All minor linguistic suggestions by reviewer 2 were revised accordingly in the manuscript. The change traces were marked with yellow color.
